# Experiences of Female Childhood Cancer Patients and Survivors Regarding Information and Counselling on Gonadotoxicity Risk and Fertility Preservation at Diagnosis: A Systematic Review

**DOI:** 10.3390/cancers15071946

**Published:** 2023-03-23

**Authors:** Nikita H. Z. Clasen, M. E. Madeleine van der Perk, Sebastian J. C. M. M. Neggers, Annelies M. E. Bos, Marry M. van den Heuvel-Eibrink

**Affiliations:** 1Princess Máxima Center for Pediatric Oncology, 3584CS Utrecht, The Netherlands; 2Faculty of Medicine, Vrije Universiteit Amsterdam, 1081CX Amsterdam, The Netherlands; 3Department of Internal Medicine, Section Endocrinology, Erasmus Medical Center, 3015GD Rotterdam, The Netherlands; 4Division of Child Health, Wilhelmina Children’s Hospital, University Medical Center Utrecht, 3584EA Utrecht, The Netherlands

**Keywords:** childhood cancer, gonadal damage, infertility, counselling, female, experience, ovarian insufficiency

## Abstract

**Simple Summary:**

Due to the significant increase in overall survival rates of childhood cancer, more awareness has been raised for the long-term consequences of treatment, including infertility. Currently, patients and their families are offered information regarding the risk of gonadal damage by paediatric oncologists and fertility counselling by fertility specialists regarding fertility preservation. However, the experiences of childhood cancer patients with oncofertility care are underreported. The available evidence reported in this review shows that female patients and survivors are variably satisfied with fertility information and report challenges in communication. They prefer to receive general information at diagnosis and detailed information later. Regrets are reported after refusal of fertility preservation. Patients and survivors are concerned about future children’s health, effect on relationships and lack of control over fertility. With the results from this review, (international) standards for information for paediatric cancer patients and families may be developed to improve fertility information and counselling for current and future childhood cancer patients and survivors.

**Abstract:**

Background: Childhood cancer patients and their families are increasingly offered oncofertility care including information regarding their risk of gonadal damage by paediatric oncologists, fertility counselling by fertility specialists and fertility preservation options. However, experiences regarding oncofertility care are underreported. We aimed to summarize the available evidence of experiences of female childhood cancer patients and survivors regarding oncofertility care. Methods: Manuscripts were systematically identified using the PubMed and Embase database. From, respectively, 1256 and 3857 manuscripts, 7 articles were included and assessed, including risk of bias assessment. Outcome measures included data describing experiences of female childhood cancer patients and survivors, regarding fertility information, counselling and/or preservation. Results: Female patients and survivors are variably satisfied with fertility information, report challenges in communication with healthcare professionals and prefer to receive general information at diagnosis and detailed fertility information later. Regrets after fertility counselling are underreported, but are associated with refusing fertility preservation. Lastly, regardless of counselling, female patients and survivors report fertility concerns about their future children’s health and effect on relationships. Conclusion: Currently, the satisfaction with oncofertility care varies and female patients or survivors report regrets and concerns regardless of receiving fertility information or counselling. These results may help to improve the content of fertility information, communication skills of healthcare professionals and timing of counselling.

## 1. Introduction

Over the past five decades, overall survival rates of childhood cancer significantly increased up to 80% in developed countries [1,2]. Currently, over 500,000 childhood cancer survivors (CCS) are alive in Europe [3]. This increasing absolute number of survivors has raised awareness for long-term consequences of childhood cancer treatment. CCS reveal a significantly increased risk of chronic health conditions in comparison to healthy peers, including secondary malignancies, heart disease and gonadal damage [4]. The risk of premature ovarian insufficiency (POI), defined as amenorrhea before the age of 40 years, after childhood cancer depends on many factors, including administered treatment and age at diagnosis [5,6,7,8,9]. Known risk factors for gonadal damage include increasing doses of abdominal or pelvic radiation (*p* < 0.001) and/or alkylating agent chemotherapy (*p* < 0.0001) [10]. Furthermore, a younger age at diagnosis is associated with a lower risk of infertility (*p* ≤ 0.0001) [5,6,7,8,9,10]. 

It is considered important to timely identify, triage and inform patients of their risk of gonadal damage early after diagnosis [11,12]. The American Society of Clinical Oncology recommends discussing fertility information with all paediatric cancer patients, even if gonadal damage risk is low [13]. Preferably, counselling by fertility experts and fertility preservation is offered to patients at risk of treatment-related gonadotoxicity as standard care [14]. The implementation of oncofertility programmes in a paediatric oncology setting has proven to be valuable to provide such timely oncofertility care [12]. Fertility information initiatives prior to cancer treatment intend to inform all patients and families on the patient’s expected gonadal damage risk by their oncologist. In addition to this, offering counselling regarding the available fertility preservation options by a fertility specialist and facilitating fertility preservation procedures for patients at (high) risk has become standard in many paediatric cancer centres. Counselling patients at high risk of treatment-related POI enables the use of fertility preservation options, while informing families of patients with a low risk of POI provides reassurance that this may not be necessary [15]. Fertility preservation options for girls include ovarian tissue cryopreservation (OTC), oocyte cryopreservation (before the start of treatment or one year after cessation of treatment in patients >15 years), oophoropexy and embryo cryopreservation. OTC is the only option for prepubertal girls or when gonadotoxic treatment needs to be started without delay. Oocyte cryopreservation can be performed in pubertal and post-pubertal patients with the time to delay oncologic treatment [16]. Embryo cryopreservation obviously needs a male partner or donor and thus is rarely applicable in the paediatric cancer setting.

To date, only a few studies have reported the experiences of female childhood cancer patients or survivors with fertility information, fertility expert counselling alongside diagnosis and/or fertility-related regrets or concerns regarding the decision to pursue or decline fertility preservation. Exploring these experiences is essential to recognize patterns, and to improve patient-centred general information and fertility counselling [17]. This may guide paediatric oncology specialists and fertility experts to enhance support for future patients [18] and increase the quality of shared decision-making [19]. This systematic review aims to identify and summarize the available evidence regarding experiences of female childhood cancer patients and survivors with fertility care at diagnosis, during or after the end of treatment. What has been reported about experience and satisfaction with oncofertility information and counselling? Which regrets regarding fertility information, counselling and preservation and concerns regarding patients’ fertility impairment have been published?

## 2. Methods

### 2.1. Search Strategy

The initial search was conducted on 17 January 2022. As great accomplishments have been made regarding childhood cancer treatment, survival and fertility counselling and preservation in the past 25 years [1,2,20,21,22,23,24,25], only relevant articles regarding qualitative and quantitative studies published in English after 1997 were included by systematically searching the PubMed database. Key words included childhood cancer, infertility, counselling and experience. The Medical Subject Heading (MeSH), Title/Abstract (TiAb) and author’s keyword (kW) terms used, including the search syntax, are provided in Appendix A. This review has not been registered.

We identified 1159 articles, of which 46 abstracts were selected for full-text screening using Rayyan (Figure 1) [26]. A cross reference check identified 12 additional articles for full-text screening. Two authors (N.H.Z.C., M.E.M.v.d.P.) independently reviewed the identified articles, including full-text screening. Disagreements between the two reviewers were resolved by consensus or by consulting a third reviewer (M.M.v.d.H.-E.). After full-text screening, 7 articles met the inclusion criteria [13,27,28,29] and were included in this review (Figure 1). 

The search was updated on 4 March 2023, using the PubMed and Embase database and using the initial search in PubMed in Appendix A. The Emtree (Exp), Title/Abstract (TiAb) and author’s keyword (kW) terms used for the Embase search, including the search syntax, are provided in Appendix A. 

With the updated PubMed search, 97 additional articles were identified, of which 45 abstracts were selected for full-text screening (Figure 1). Using the Embase database, we identified 3857 articles, of which 34 abstracts were selected for full-text screening (Figure 1). The search update did not yield any additional studies. 

### 2.2. Inclusion and Exclusion Criteria

Published manuscripts were included if they described female childhood cancer patients or survivors and their experiences (satisfaction, regrets and/or concerns) after fertility information, counselling or preservation at diagnosis, during or after end of treatment in a paediatric cancer setting (Table 1). Childhood cancer patients are defined as children (age 0–18 years old) diagnosed with and/or treated for childhood cancer. Childhood cancer survivors are defined as children (age 0–18 years at diagnosis) who have completed their cancer treatment at the time of study [31]. Since paediatric cancer also occurs in patients aged 18 years and older and generally the same oncologic and fertility care is provided in this group, articles describing paediatric cancer patients diagnosed between the ages of 0 up to and including 18 years were included in this review. Studies were included if ≥75% of the cohort was female or if female-specific results were provided (Table 1) [32]. Articles describing the experiences of exclusively males, healthcare professionals or parents were excluded. 

### 2.3. Data Extraction

From the included studies, the study characteristics, including study design, patient demographics and tumour types, age at study and at diagnosis, data collection and outcome measures were extracted. Outcomes are categorized in (1) experiences and satisfaction with oncofertility information and counselling, (2) regrets after fertility information and counselling and (3) concerns regarding patients’ fertility impairment, after fertility information, counselling or preservation. Fertility information is considered counselling when information is given by a fertility expert, such as a gynaecologist. Fertility information given by any other healthcare professional, such as a paediatric oncologist or nurse practitioner, is not considered counselling [12]. The quality of the studies was assessed based on their risk of bias using the seven criteria for *Risk assessment of bias in non-intervention studies and qualitative research* [34,35], including theoretical framework, aims and objectives, description of context, sample and methodology, analysis of data and sufficient original data. According to the risk of bias criteria, a study was classified as having a low risk of bias if it met four or more criteria—below three was considered a high risk of bias.

## 3. Results

Four included studies reported data on both female and male cancer patients and/or survivors [13,27,28,29] (Table 2, Appendix A). Three included studies reported female-specific data [36,37,38] (Table 2, Appendix A). The cohorts of the included studies consisted of only patients (*n* = 19; 6 months–3 years after diagnosis) [38], both patients and survivors (*n* = 17; 3 months–14 years after diagnosis) [28] and survivors only (*n* = 110), defined as being off therapy for a variable time (0–2 months to >2 years) or >5 years after diagnosis [13,27,29,36,37]. The age at diagnosis or at the time of study or tumour type was not always reported for the female participants specifically. The mean/median age of the participants ranged from 12 to 33 years at the time of study, and the mean/median age at diagnosis ranged from 5.25 to 16.5 (range 0–18 years–<12 years could not be extracted from the articles) [13,27,28,29,36,37,38]. The selected studies included patients with leukaemia, lymphomas [13,27,28,36,37,38], sarcomas [27,28,37], germ cell tumours [27], central nervous system tumour [13,27,36,38], adrenal tumours [13], renal tumours [36], rhabdomyosarcomas [28], bone tumours [13,28,36] and soft tissue tumours [37]. Participants’ risk of gonadal damage was reported by one included article, stratified into low risk (LR; 3/8 female, 38%), medium risk (MR; 10/13 female, 77%) and high risk (HR 4/16 female, 25%) groups, based on diagnosis and received therapy (Table 3) [28]. Only the females from the MR group were included, in accordance with the inclusion criteria (Table 3) [28]. Furthermore, five patients believed they had a low risk of gonadal damage, although this was not clinically confirmed [27,29,38]. Fertility information or counselling was provided by healthcare professionals or physicians, not further specified [13,37], or specifically paediatric oncologists, fertility experts or nurses [27,28,36,38]. Other reported sources of fertility information were family [13,37,38], friends or romantic partners [37], and own research by participants [37]. Wright et al. did not report from what source participants received fertility information [29]. Only two of the seven studies specified that fertility information/counselling was given at diagnosis (before/after start treatment not specified) or at the end of treatment [27,36]. 

### 3.1. Experience and Satisfaction with Oncofertility Information and Counselling

For 75 of the 146 eligible female patients and survivors, it was reported that they recalled receiving fertility information, and 5 recalled that they had been counselled at diagnosis by a fertility expert [36] (Table 3). However, the subgroups receiving information or counselling by a fertility expert could not be distinguished in all included articles. In the selected seven studies, participants’ satisfaction with the provided information on predicted reproductive health ranged from 4% to 91% (Table 4) [28,29,36,37,38], and after counselling (when specified), from 66% to 80% [36]. Oosterhuis et al. did not report satisfaction results for the risk groups separately; thus, female-specific data could not be extracted regarding satisfaction [28]. The majority of experiences with fertility information or counselling was not satisfactory [13,27,28,29,36,37]. The oncofertility programme described by Zarnegar et al. reports higher satisfaction (10/11 [91%], 1/11 missing data) in the subgroup who recalled a discussion about fertility compared to the subgroup who did not recall that fertility was discussed, in which 5/8 (63%) were satisfied with their fertility knowledge [38]. Kim et al. separately reported experiences receiving fertility information from a healthcare professional (not otherwise specified) or counselling by a fertility specialist both at diagnosis and after the end of treatment (Table 3) [36]. Of 56 female CCS (unknown gonadal damage risk status), 31 (55%) received fertility information at diagnosis, of which 5 (9%) experienced counselling by a fertility expert and 31 (55%) were informed after treatment completion, of which 3 (5%) received counselling (55%). Of the 51 women not counselled by a fertility expert, 21 (41%) would have appreciated to be counselled at diagnosis [36]. A total of 40/56 women (71%) were not satisfied with the received information at diagnosis compared to 34/56 (61%) after treatment and 40/56 women (71%) had or would have pursued fertility counselling by a fertility expert [36]. Four out of five (80%) and two out of three (67%) receiving counselling from a fertility specialist at diagnosis or end of treatment, respectively, were satisfied (Table 3) [36].

The majority of female childhood cancer patients and survivors (61–96%) preferred more information on the effect of cancer treatment on their fertility at an early stage [13,27,28,29,36,37] and the preferred time for information about fertility matters suggested by patients (*n* = NR) was at diagnosis [27,36]. Patients and survivors (*n* = NR) welcomed the information, regardless of gonadal damage risk or fertility preservation options [19], some (*n* = NR) even reported they believed that the offer of fertility preservation was an expression of professional belief that they had a future [27]. Participants (*n* = NR) preferred ‘broad’ information around diagnosis, provided that it was age-appropriate and participants (*n* = NR) appreciated the opportunity to ask questions and have a more detailed discussion later, if needed [27]. Furthermore, some participants (*n* = NR) indicated that they would have found it helpful if information was offered repeatedly and in an unambiguous, low key manner [27]. Participants (*n* = NR) appreciated the involvement of only a small number of professionals and family members in fertility discussions [27]. 

Reported reasons for dissatisfaction included problems in communication with healthcare professionals; information provided in an insensitive manner; not revisiting the topic of fertility and insufficient, incorrect, unclear or confusing information regarding participants’ expected future fertility status or fertility preservation options [13,27,29]. Wright et al. reported that three out of four women diagnosed under 18 felt that the information regarding fertility was inadequate [29]. Crawshaw et al. reported that women (*n* = NR) had difficulty understanding fertility information, even after they had received counselling and regardless of whether they accepted or refused fertility preservation options [27]. 

### 3.2. Regrets Regarding Fertility Information and Counselling

Regrets of female CCS after fertility counselling were underreported in the literature (Table 4). Only Crawshaw et al., (2009) reported findings on regrets in a cohort of 21 female CCS (aged 11–18 years) after fertility information and counselling, which were all related to the refusal of fertility preservation [27]. Fertility preservation had been offered to 5 out of 21 women (24%) with a high risk of gonadal damage: 3 (14%) were offered oocyte cryopreservation, 1 (5%) was offered a procedure to protect her ovaries during radiotherapy and 1 (5%) had her ovaries ‘tied up’, but she did not understand what this meant [27]. The four women who had refused fertility preservation mainly decided this to avoid delaying cancer treatment and were grateful to have had the offer. However, years later after facing impaired fertility (*n* = 3) or uncertainty regarding fertility status, they contemplated the circumstances of the offer made, including timing of the offer, and questioned their decision [27]. It was described as “times of later preoccupation with the circumstances of their decision” [27] (p. 385) and they “questioned the wisdom of the offer being made and felt that this made their later (unexpected) infertility more difficult to cope with” [27] (p. 385). No regrets on accepting fertility preservation were reported. 

### 3.3. Concerns Regarding Fertility Impairment

Four studies reported on concerns childhood cancer patients and survivors have related to the expected fertility-related side effects of cancer treatment (Table 4) [13,28,36,37]. Concerns related to fertility information or counselling were not reported in the included articles. Fertility-related concerns of female participants were consistent among studies—these include general concerns regarding impaired fertility, which often persist even after information has been provided (Table 4) [13,28,36,37]. Other concerns include risk of hormone production deficiency (2/13, 15%), impact on puberty (3/13, 23%), altered genetic material in gametes (6/13, 46%) [28] and the effect of treatment on future offspring’s health (46–60%) [28,37]. Two studies mentioned that many participants (45–>50%) would feel sad if they were unable to have a child [13,37]. For >10% of participants, decreased fertility is the most important cancer-related concern, and for >25%, fertility is among the top 3 after concerns regarding survival and risk of cancer recurrence [36]. Furthermore, Jardim et al. reported that participants (*n* = 11, 100%) had concerns regarding the effect of potential fertility damage on future romantic relationships [13]. Women (*n* = NR) feel additional distress due to the perceived need to disclose their potential infertility early in a relationship, and this distress is associated with the fear of abandonment [13]. 

## 4. Discussion

The overall survival rate of childhood cancer has improved significantly [1,2], which urges the need to pay more attention to improving the prevention of toxicities. Gonadotoxicity is a common and highly relevant toxicity as reported by several studies in CCS [4,5,6,7,8,9]. This systematic review highlights that the experiences of childhood cancer patients and survivors concerning fertility counselling before, during and after treatment have hardly been addressed so far [15,19]. Overall, the results show that female CCS are often dissatisfied with the provided fertility information and the communication about fertility with their healthcare professionals [13,27,28,29,36,37,38] compared to better experiences after fertility counselling by a fertility expert [36]. Regrets were only reported in one study (*n* = 21), and seem to be associated with refusal of fertility preservation [27]. No concerns about the counselling specifically have been reported in childhood cancer patients or survivors. Still, childhood cancer patients and survivors are concerned about their future children’s health and the effect on romantic relationships [13,28,36,37]. 

While the results remain consistent between most studies, the availability of and differences between oncofertility programmes [29,37], a younger age and whether information is given to the patient by the parents, healthcare professionals, paediatric oncologists or fertility experts may have an effect on experience and satisfaction [13,16,37]. The American Society of Clinical Oncology recommends discussing fertility information with all paediatric cancer patients, even if gonadal damage risk is low [13]. When this is not feasible, information can be given to the parents first [13]. However, it is recommended that a healthcare professional initiates fertility discussion with patients later in life [13]. In the study by Wilkes et al. (*n* = 18; including two female CCS—aged 4 and 5 years at diagnosis), suggestions of CCS to improve fertility care included providing an informative leaflet or web-based information and access to fertility specialists, giving information as early as possible, but also at various stages during treatment and subsequent check-ups allowing patients/survivors the opportunity to discuss fertility at different stages of their lives [39]. Ellis et al. reported that some survivors (*n* = 3) indicated that they would have liked to know more about their fertility status and whether or not they might be able to have children [40]. In line with the findings in this review, there was a general feeling that more information could have been offered about fertility, available fertility preservation options and associated risks [39,40].

### 4.1. Experience and Satisfaction with Oncofertility Information and Counselling

Dissatisfaction with fertility care was most often related to poor communication and/or incorrect, confusing, unclear or insufficient information. The relatively complex information regarding fertility and fertility preservation in women has been reported to be confusing and leading to difficulty understanding the information (*n* = NR) [27]. Besides, the general knowledge about fertility including the association between the menstrual cycle and fertility was not always clear [13,27,36,37]; hence, education is important during fertility information and counselling. Additionally, the timing of the information was an important factor, as the stressful diagnostic period can lead to inadequate information perception, misunderstanding and confusion [27]. Ellis et al. [40] reported similar findings regarding poor communication in a combined cohort of female and male CCS (*n* = 19) and their parents: three survivors reported challenges in communication with professionals, resulting in uncertainty regarding their future fertility [30]. 

### 4.2. Regrets Regarding Fertility Information and Counselling

Information on regrets in childhood cancer patients and survivors with and after fertility counselling is hardly available in the published literature. As regrets were only discussed in one article with a relatively small sample (*n* = 17) [27], regrets after fertility counselling and fertility preservation may be further addressed in future studies to improve fertility counselling (Table 4). In line with our findings, Bastings et al. [17] reported findings on decisional conflict (difficulty in decision-making) and decisional regret (current regrets regarding past decisions) regarding fertility preservation in patients older than 16 years (*n* = 60). Decisional regret is shown to be correlated with decisional conflict (*p* < 0.0001) [17]. Factors influencing decisional conflict were not enough time for counselling, not having the opportunity to ask all questions during counselling, not feeling supported by healthcare professionals during decision-making, not all applicable options were discussed and the benefits and disadvantages of fertility preservation were not clearly explained [17].

### 4.3. Concerns Regarding Fertility Impairment 

Fertility-related concerns reported by female childhood cancer patients and survivors include the risk of infertility (8/13, 62%), entering menopause prematurely, impact on the normal progression of puberty (3/13, 23%) [28], effect of cancer treatment on genetic material and gametes (46–60%) [28,37] and the effect of potential infertility on romantic relationships (*n* = 11, 100%) [13]. Crawshaw et al., (2010) [41] reported that female CCS (*n* = NR) experienced distress and frustration due to the (perceived) lack of control over their impaired fertility [41]. These concerns are described to become more prominent after the intensive treatment has ended [40]. Similar studies in female childhood cancer patients and survivors (*n* = NR) report concerns both when a child wish occurs [39,41], but also when survivors were not planning to start a family for a considerable amount of time [40]. These concerns seem to become more prominent when patients have been insufficiently informed, the fertility counselling has been unclear or the timing of counselling was inappropriate [39,40,41]. In most studies, information on the potential risk of gonadal damage was provided at diagnosis. Since the diagnostic period is a hectic and emotional time, patients can easily forget certain information, possibly leading to future fertility-related uncertainty and concerns. Therefore, postponing detailed information for days or weeks may be considered, whenever the clinical situation could allow for this [12].

### 4.4. Strengths and Limitations and Recommendations for Future Research

This systematic review provides the first analysis of a subgroup of the population, female CCS. Male [42] or (young) adult female [43] cancer patients and survivors’ experiences with and after fertility information and/or counselling have been published. Few manuscripts report these experiences of female childhood cancer patients and survivors. There is a considerable difference between male and female fertility counselling and preservation: females need more complex information regarding fertility preservation options, which involve surgery. Therefore, male experiences cannot be used for females, and more research may be conducted in female CCS. Previously published systematic reviews regarding fertility counselling and/or preservation have either included only adults, focused on childhood cancer patients and/or survivors without providing female-specific data, or focused on experiences with fertility preservation, without taking experiences with information and counselling into account. None of these reviews focused specifically on the experiences of female childhood cancer patients and/or survivors with fertility information and/or counselling during and after childhood cancer. Hence, this review provides valuable insights regarding female paediatric oncofertility care and the existing gaps of knowledge.

Some gaps of knowledge still remain regarding female experiences of fertility counselling. Few studies have studied the experiences and perspectives regarding fertility care of female childhood cancer patients or survivors specifically. Many identified articles (*n* = 43) included ≤75% female participants or participants diagnosed after 18 years, or did not provide a sub-analysis of female participants. In paediatric oncology, patients older than 18 are also treated according to paediatric treatment protocols. Therefore, patients aged 18 years were also included in this review. Due to the qualitative nature of several articles, these articles may have been subjectively biased and statements of included participants may not be generalizable and may differ from participants from other age groups or countries. We expect the results to vary between different risk groups (low, medium and high). Only one study in this review stratified between risk groups [28]. Three articles reported that participants (*n* = 5) believed that they had a low risk of gonadal damage, but this was not confirmed or further specified [27,29,38]. Multiple studies interviewed participants retrospectively, representing patients’ and participants’ experiences well and leading to better generalisability; however, this could also lead to recall bias [36,38]. Since this review focusses on the experiences of female childhood cancer patients and survivors specifically, the experiences of parents with or after fertility counselling were not included in this review. We anticipate that the experiences of children informed at a later age by their parents vary considerably from the experiences of patients or survivors who were directly informed by a healthcare professional. However, since parents make the majority of the decisions concerning cancer treatment for patients aged under 12 years, it is also important to evaluate the experiences of this group. It may be valuable for future research to focus on the concerns and regrets of patients under the age of 12 years and who are primarily informed by their parents, thus receiving second-hand information.

Regrettably, due to the limited available published studies, evidence-based recommendations cannot be provided. Our recommendations regarding future research are based on these remaining gaps of knowledge. We believe it is valuable to conduct more research on the experiences of female childhood cancer patients and survivors with or after fertility information and/or counselling. Patients’ gonadal damage risk and whether they received first- or second-hand information need to be taken into account. It may also be valuable to study the experiences of parents of female childhood cancer patients and survivors specifically, also taking into account their daughter’s risk of gonadal damage. 

This systematic review highlights the importance of optimizing and personalizing oncofertility care by enhancing the content of comprehensive information early in childhood cancer treatment at a convenient moment for newly diagnosed female childhood cancer patients and to repeat this information overtime. It confirms the importance of informing all patients and families (also those at low risk of gonadal damage) and providing counselling by fertility experts in higher risk patients to ensure that oncofertility care is tailored to patients, regarding their age, gender, wishes and needs [11,13]. These recommendations have been implemented and evaluated in a national cohort in the Princess Máxima Center for Pediatric Oncology since 2018 [12]. The findings of this study confirm that uncertainty regarding fertility status is still an important concern [27,37]. 

In conclusion, childhood cancer patients and survivors are variably satisfied with oncofertility care. Improvements can be made in providing personalized fertility risk information, repeating this information during treatment and survivorship, improving communication by the physician and stimulating fertility counselling by fertility experts. Regrets after declining fertility preservation have been reported as well as fertility-related concerns, regardless of counselling, including concerns for the health of future offspring. Larger studies are needed, reflecting on the experiences, concerns, regrets and satisfaction regarding fertility information, counselling and/or preservation, to improve the quality of oncofertility care.

## Figures and Tables

**Figure 1 cancers-15-01946-f001:**
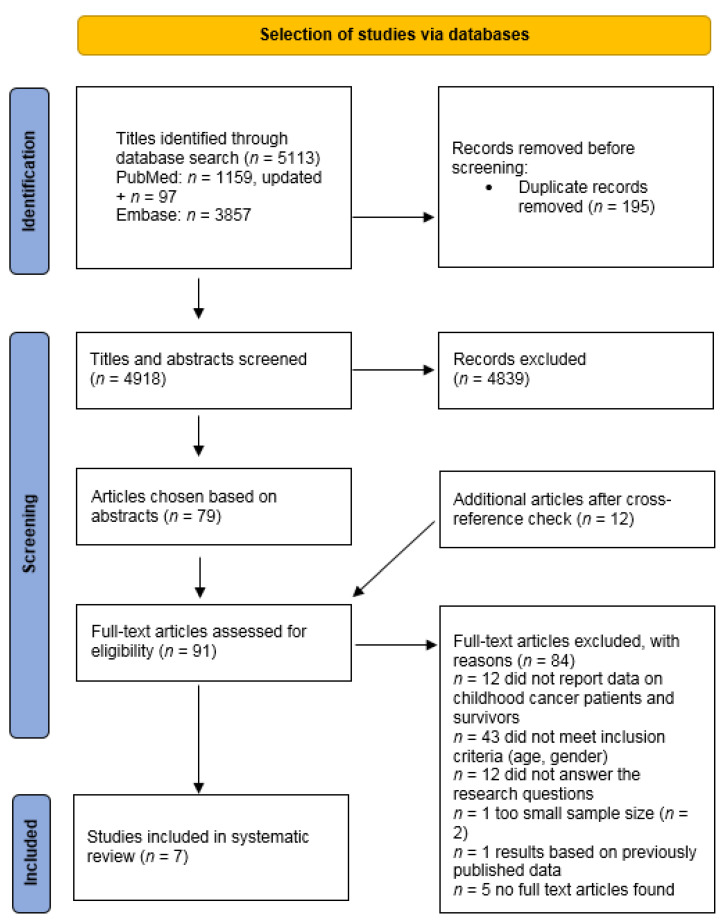
PRISMA study flow diagram of the search strategy in the PubMed and Embase Database, updated on 4 March 2023 [30].

**Table 1 cancers-15-01946-t001:** Inclusion and exclusion criteria for studies using the PICOTSS Framework [32,33].

PICOTSS	Inclusion	Exclusion
Population	Female patients or survivors of childhood cancer ≥75% of the population diagnosed between the ages of 0–18 years or a sub-analysis of participants ≤18 years ≥75% of the population female or a sub-analysis of female participants	Male patients or survivors of childhood cancer Patients or survivors diagnosed with (young) adult cancer, such as gynaecologic or breast cancer Primary focus on parents of childhood cancer patients or survivors Primary focus on healthcare professionals or specialists
Intervention	Fertility information or counselling	Lack of focus on infertility, fertility information or counselling
Comparator	Not applicable for our study	Not applicable for this study
Outcomes	Experiences, including satisfaction, regrets and/or concerns	Lack of focus on experiences Experiences of parents, healthcare professionals or specialists of childhood cancer patients or survivors
Timing	At childhood cancer diagnosis, during or after end of treatment	Not applicable for our study
Setting	Paediatric cancer setting	(Young) adult cancer setting
Study design/ other limiters	Articles written in English after 1997	Articles not written in English and/or before 1997 Reviews, systematic reviews, narrative reviews, literature reviews, short communications, guidelines, case reports, case series

**Table 2 cancers-15-01946-t002:** Summary of patient characteristics and risk of bias assessment of included studies.

Reference	Design of the Study	Eligible Participants (F;M) ^a^	Age at Diagnosis in Years Median (Range) ^b^	Participant Characteristics	Diagnoses	Risk of Bias (RoB) ^g^							RoB
						i	ii	iii	iv	v	vi	vii	
Crawshaw et al., 2009. [27]	Qualitative (grounded theory), single interviews	*N* = 13 *T* = 38 (21;17 ^c^)	15 (11–18) ^d^	Survivors	SC, LYM, LEU, GCT, CNS ^d^	(+)	(+)	(+)	(+)	(−)	(+)	(+)	L (6/7)
Jardim et al., 2020. [13]	Qualitative, SSI	*N* = 11 *T* = 24 (11;13)	8.13 (1–15) ^d^	Survivors	ST (OS, ES), LYM, LEU, CNS, AT ^d^	(+)	(+)	(+)	(+)	(+)	(+)	(+)	L (7/7)
Kim and Mersereau. 2014. [36]	Quantitative, cross-sectional, web-based survey	*N* = 56	Mean 13 (7–19)	Female survivors	LEU, LYM, CNS, RT, BT, STT, OTH	(−)	(+)	(+)	(+)	(+)	NA	(+)	L (5/7)
Oosterhuis et al., 2008. [28]	Quantitative survey	*N* = 17 *T* = 37 (17;20)	Female: NR ^e^.	Patients or survivors	LYM, LEU, NBL, RMS, OS, SC ^d^	(−)	(+)	(+)	(+)	(+)	NA	(+)	L (5/7)
Sandheinrich et al., 2018. [37]	Quantitative, cross-sectional, questionnaires	*N* = 26.	5.25 (0.68–15.68)	Female survivors	CNS, LEU, LYM, SC, ST	(−)	(+)	(+)	(+)	(+)	NA	(+)	L (5/7)
Wright et al., 2014. [29]	Qualitative, quantitative, SSI	*N* = 4 *T* = 14 (5;9) ^f^	16.5 (14–18) Total: (12–24) ^d^.	Survivors	NR	(−)	(+)	(−)	(−)	(−)	(−)	(+)	H (2/7)
Zarnegar et al., 2017. [38]	Observational pilot, online survey	*N* = 19	Mean: 15.6 (13–18)	Female patients	LYM, SC, LEU, OT, OTH, CNS	(−)	(+)	(+)	(+)	(−)	NA	(+)	L (4/7)
Total results	Survey: 4 Interviews: 3	*N* = 146	NA	Survivors and patients	SC, LYM, LEU, GCT, CNS, ST (OS, ES), AT, RT, BT, STT, NBL, RMS, OT, OTH	2	7	6	6	4	2	7	L Mean (range): 4.9 (2–7)

Results presented in this table are the results for the eligible participants according to our selection criteria, unless specified otherwise. ^a^: *T* = total number of patients included in the study. *N* = number of female participants under the age of 19 years at diagnosis. ^b^: Unless stated otherwise. ^c^: 4 females aged 19 or 20 at diagnosis. ^d^: Data not female-specific. ^e^: MR group: Mean age at study 19.4 (16–25); mean since diagnosis 62.5 months (4–129). ^f^: One female 20 years at diagnosis, 22 at study. ^g^: (+) study fulfils criteria; (−) study does not fulfil criteria or it is unknown [34,35]. i: An explicit account of theoretical framework and/or the inclusion of a literature review which outlined a rationale for the intervention. ii: Clearly stated aims and objectives. iii: A clear description of context which includes detail on factors important for interpreting results. iv: A clear description of sample. v: A clear description of methodology, including systematic data collection methods. vi: Analysis of data by more than one researcher. vii: The inclusion of sufficient original data to mediate between data and interpretation. RoB: criteria risk of bias assessment non-intervention studies; H: High risk bias, 0–3 points; L: Low risk bias, 4–7 points. SSI: semi-structured interviews. NR: Not reported. MR: Medium risk. NA: Not applicable. SC: sarcoma, including spindle cell sarcoma; LYM: lymphoma; NHL: Non-Hodgkin lymphoma; LEU: leukaemia; GCT: germ cell tumours; CNS: central nervous system tumours, including brain tumour; NBL: neuroblastoma; ST: solid tumours; RMS: rhabdomyosarcoma; AT: adrenal tumour; RT: renal tumour; BT: bone tumour; OS: osteosarcoma; ES: Ewing’s sarcoma; STT: soft tissue tumour; OT: ovarian tumour; OTH: others.

**Table 3 cancers-15-01946-t003:** Summary of experiences on obtained oncofertility information and counselling in a paediatric cancer setting.

Reference	Participant Characteristics	Gonadal Damage Risk Group (F;M)	FI: at (Time) by (Person (*n*))	Key Results		
				Experience and Satisfaction	Regrets	Concerns
Crawshaw et al., 2009 [27]	*n* = 13	NR	Recall FI: *n* = NR Most at diagnosis by PO (*n* = NR)	Most (*n* = NR) received FI. None counselled by FE. FI wanted independent of risk or FP options. Most FI (*n* = NR) not satisfactory. Unclear and confusion information about FI and FP. Not female-specific: professional or parental support wanted ^bc^.	Declined FP offer (*n* = 4). Doubts about timing offer.	NR
	Survivors aware fertility might be affected.	LR: *n* = 1 ^a^				
Jardim et al., 2020 [13]	*n* = 11	NR	Recall FI: *n* = 4	Mother informed *n* = 4.7 have an unknown fertility status.	NR	Feeling sad possible infertility (*n* = 5). No difficulty telling partners potential infertility (*n* = 11), but still have concerns, associated with fear abandonment.
	Survivors >5 years after diagnosis, aged 18–24 years.		Time: NR Parent: 4	4 have doubts about fertility. 1 miscommunication. The majority (*n* = 10/11, 91%) wants more FI.		
Kim and Mersereau. 2014 [36]	*n* = 56 Female survivors aged 18–45 years.	NR	Recall FI: *n* = 31 At diagnosis: HP: 26; FE: 5	At diagnosis ^c^: pursued FC: 4/5 (80%) satisfied. 21/51 wanted FC. 25 (44%) had no recall of FI.	NR	Decreased fertility important concern.
	5 with POI. 16 with irregular menses.		After treatment: HP: 28; FE: 3 Parents: NR	40/56 (71%) (diagnosis ^c^), and 34/56 (61%) (end of treatment) need more FI, thus 16/56 (29%) (diagnosis ^c^), and 22/56 (39%) (end of treatment) are not satisfied with FI. After treatment: 2/3 (66%) FC-satisfied		
Oosterhuis et al., 2008 [28]	*n* = 17	LR: 8 (3;5) MR: 13 (10;3)	Recall FI: *n* = NR Time: NR	Female-specific: NR; risk-group-specific: NR (similar between risk groups LR, MR and HR).	NR	MR: fertility-related concerns; 8/13 infertility, 2/13 hormone production, 3/13 impact on puberty, 6/13 effects on genetic material of gametes. LR and HR group female group too small ^b^
	Patients or survivors (>14 years) (routine treatment/ follow-up).	HR: 16 (4;12)	HP (incl PO, doctors, nurses)	13/37 (35%) satisfied with provided FI ^b^. 14/37 received educational material ^b^, 8/37 received FP information ^b^, 17/37 want more FI (from FE/PO/nurse) ^b^.		
Sandheinrich et al., 2018 [37]	*n* = 26	NR	Recall FI: *n* = 25 Time: NR Physician (not specified): 9;	96% of participants wants more FI, thus 4% satisfied. 69% frustrated about risk of infertility. Only 27% felt they had control over their fertility.	NR	>50% survivors sad if infertile. A total of 60% concerns future child’s health, >20% cancer recurrence.
	Female survivors aged 13–18 years (no documented infertility).		Parent: 11; Peers: 1; Own research: 4			
Wright et al., 2014 [29]	*n* = 4 Survivors 2 months–4 years after treatment.	NR	Recall FI: *n* = 4	3/4 females informed. 1/4 (25%) well informed (but unclear fertility status). Mixed levels of received FI: inadequate, insensitive. The majority (*n* = 3/4) are unaware of their fertility status. Unclear where to obtain more FI (*n* = 2).	NR	NR
		LR: *n* = 1 ^a^	Time: NR Person: NR			
Zarnegar et al., 2017 [38]	*n* = 19	NR	Recall FI: *n* = 11; chart: *n* = 10 Time: NR	11/19 (58%) recall discussion infertility, 8/11 initiated by doctors, nurses, psychologists, social workers, family (*n* = NR). 9/19 (47%) discussed FP (1 OTC, 2 HormT, 6 declined). 10/11 (91%) satisfied with FI (1/11, missing data) ^d^. 5/8 without FI are satisfied with fertility knowledge.	NR	NR
	Female patients 6 months–3 years after diagnosis.	LR: *n* = 3 ^a^	Doctors, nurses, psychologists, social workers, family (*n* = NR)			
Total results	*n* = 146	NA	Informed (incl parent): >75 Counselled: 5	Variable satisfaction 4–91%. After counselling satisfaction 66–80%.	Regrets declining FP.	Regarding own fertility and future offspring.
	Survivors and patients.					

^a^: Participants believed they had a low risk of gonadal damage; however, authors did not report whether this was really the case. ^b^: Data not female-specific. ^c^: Fertility counselling received at diagnosis; however, unclear whether before or after start treatment. ^d^: One participant recalls discussion on fertility, but did not answer all (satisfaction) questions, unclear why. NR: Not reported. n: number of female participants under the age of 19 years at diagnosis. RC: reproductive care; FI: fertility information; FC: fertility counselling; FP: fertility preservation; PO = paediatric oncologist; FE = fertility expert; HP: healthcare professional (e.g., PO, FE, nurses, doctors); HormT = hormonal therapy.

**Table 4 cancers-15-01946-t004:** Important findings of included studies.

Important Findings
Satisfaction
High variation in satisfaction amongst studies (4–91%);
Participants preferred information on the effect of cancer treatment on fertility, to be informed at diagnosis, to receive broad information at first and later detailed information;
Participants preferred involvement of a small number of professional and family members;
Participants pleased to be offered the opportunity for fertility preservation;
Dissatisfaction amongst participants occurred when there were problems: in communicating with healthcare professionals: information was provided insensitively or downplaying of the importance of fertility matters; with the received in formation: information was insufficient regarding fertility and participants prefer better additional written fertility information.
Concerns
Consistent and repetitive amongst included studies;
Concerns remained present, even after fertility information was received;
Fertility-related concerns of great importance to female patients and CCS;
Fertility-related concerns have an effect on participants’ future career, life, parenthood and relationships.
Regrets
Underreported in included studies;
When reported, the authors were unclear as to what they meant.

## Data Availability

All data is presented in the manuscript and Appendix A.

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
