# Peer review of "Experiences of Female Childhood Cancer Patients and Survivors Regarding Information and Counselling on Gonadotoxicity Risk and Fertility Preservation at Diagnosis: A Systematic Review"

_cancers, 2023, doi:10.3390/cancers15071946_

Round 1
Reviewer 1 Report
This manuscript summarizes the experiences of female childhood cancer patients and survivors regarding oncofertility care.
Abstract
- The simple summary states “They prefer to receive general information at diagnosis and detailed information later.” However, the abstract states, “female patients… prefer to receive information at diagnosis.” Consider including the statement in the simple summary in the abstract, as this is an important contribution to the literature and is missing a key piece of information as it reads in the abstract now.
Background
- Line 55: Direct reference to clinical guidelines to anchor readers on evidence-based recommendations would be helpful here. This is done in the discussion, lines 276-280, and can remain here, but should be included as part of introducing the review as well.
- Last paragraph: Consider including guiding questions following the aims of the review. This may help organize findings (e.g., similar to the sections in the results [3.1. Experience and satisfaction with oncofertility information and counselling]) throughout the review.
Methods
- Consider using the Population, Independent variables/intervention, Comparator, Outcomes, Timing, Setting, and Study design/other limiters (PICOTSS) framework to comprehensively report reference inclusion/exclusion criteria.
- Section 2.2: There is no mention of treatment status as it pertains to inclusion or exclusion in this review. Building on this, as both “patient” and “survivor” terminology is used, it is important to operationally define what is meant by each for the purposes of this review.
- Details pertaining to reference screening and literature review are missing. Information such as how many reviewers were included at each stage, how many reviewers screened each reference, how conflicts were resolved, etc. should be included.
- The PRISMA study flow diagram should be referenced and cited accordingly.
Results
- Lines 140-152: Information presented in the first paragraph of the results section pertaining to risk of infertility and counseling, among others, is hard to follow in its current format. Consider reorganizing this in a similar fashion to Table 2.
Discussion
- Consider including subheadings in the discussion, consistent with guiding questions developed for the introduction and the subheadings used in the results section, to help with organization. In addition, a strengths and limitations section would be helpful.
- A detailed future directions section to synthesize the findings and make evidence-based recommendations for missing research would be invaluable.
Thank you for the opportunity to review this manuscript.
Reviewer 2 Report
Thank you for the opportunity to evaluate this review manuscript entitled: Experiences of Female Childhood Cancer Patients and Survivors regarding Information and Counselling on Gonadotoxicity. Risk and Fertility Preservation at Diagnosis.
Despite oncofertility is already an important clinical topic, this review is based only on very few publications. Originally identified were 1159 articles, but after full-text screening, 7 articles met the inclusion criteria and were included.
The analysis and reporting of these 7 articles is well performed and comprehensive.
It was shown that childhood cancer patients and survivors are variably satisfied with oncofertility care. Patients and survivors report regrets and concerns regardless of receiving fertility information or counselling.
Thus, improvements can be made in providing personalized fertility risk information, repeat this information during treatment and survivorship, and improving communication by the physician and stimulate fertility counselling by fertility experts.
The conclusions are driven as well as they could be from the scarce material available.
It is imminent that larger studies reflecting on the experiences, concerns, regrets and satisfaction regarding fertility information, counselling and preservation would increase needed knowledge.
Reviewer 3 Report
In their review “Experiences of Female Childhood Cancer Patients and Survivors regarding Information and Counselling on Gonadotoxicity Risk and Fertility Preservation at Diagnosis: A Systematic Review”, authors emphasize the fact that, due to the increase in survival rates of childhood cancer, more awareness is raised for long-term consequences of infertility. However, the experiences of childhood cancer patients with oncofertility care are underreported. Patients and survivors are concerned about future children’s health, effect on relationships and lack of control over fertility. The present work is focused on the few scientific articles that specifically address this topic.
The state of the art carried out in this review is very well written, the writing is neat, the methodology used is very clear (and precise) and the articles that came out - few in number - are very well studied. However, in order to improve the review, some small modifications or remarks need to be made:
1. Make section breaks between text and tables while maintaining continuity of page numbering (indeed, pages return to “1” several times). This is valid because the journal cancers in the MDPI group requires authors to do the near final layout.
2. Page 1, line 13. Please, check that “Short title:” should not be entirely in bold
3. Page 1, line 39. Please, add a space between “relationships.” and “:”.
4. Page 1, line 43. Could the authors think about finding keywords that may be more relevant? Does the keyword “experience” really seem to be related to this review? Also, is it possible to add a keyword related to ‘female’, ‘girls’ or ‘woman’ and for ‘ovarian insufficiency’?
5. Page 2, line 90. Please, check that the reference 18 “Oktay K, Harvey BE, Partridge AH, Quinn GP, Reinecke J, Taylor HS, et al. Fertility Preservation in Patients With Cancer: 424 ASCO Clinical Practice Guideline Update. J Clin Oncol. 2018;36(19):1994-2001” published in 2018 (and referring to publications of ASCO since 2006) is a good reference to justify 25 years of great accomplishments regarding childhood cancer treatment.
6. Page 4, lines 127 to 128. Please, check that the “Supplemental tables S2-S8” are well correlated with “Four included studies reported data on both female and male cancer patients and/or 127 survivors (22-25)”. Idem for the sentence “Three included studies reported 128 female-specific data (26-28)” without reference of Supplemental tables”.
7. Page 4, line 136. Please, rearrange the sentence “(range 0-18 years (<12 years could not be extracted from the articles))” so that there are no more double brackets (e.g., use brackets and/or dashes).
8. Table 1, page “2”, top right. Please, add a space between “(range):” and “4.9”.
9. Table 1, page “2”, in the middle at the bottom. Please, suppress the space between “wanted.” and “bc”.
10. Page “1?”, line 187. Please, rearrange the sentence “(10/11 (91%), 1/11 missing data)” so that there are no more double brackets (e.g., use brackets and/or dashes).
11. Page “1?”, lines 190 to 191. Please, rearrange the sentence “(not otherwise specified (NOS))” so that there are no more double brackets (e.g., use brackets, comma or semicolon).
12. Page “3?”, line 238. Please, check that the reference “[22](p. 385)” is not a typo (or a leftover from a bibliography system).
13. Page “3?”, line 240. Please, check that the reference “[22](p. 385)” is not a typo (or a leftover from a bibliography system).
14. Page “3?”, lines 281 to 282. Please, rearrange the sentence (aged 4 and 5 years at diagnosis))” so that there are no more double brackets (e.g., use brackets and/or dashes).
Reviewer 4 Report
In this systematic review, the authors identified and summarized the available experiences of female childhood cancer patients and survivors regarding fertility information and counselling during and after childhood cancer focusing on satisfaction with the received oncofertility care, regrets and concerns. They found that the satisfaction with oncofertility care varies and female patients or survivors report regrets and concerns regardless of receiving fertility information or counselling. This is a study with a certain significance, but some points are required to be improved.
1.There were some systematic review have investigated the experiences and perspectives regarding fertility care and concerns in female childhood cancer patients or survivors in recent years. What‘s the differences between this manuscript and others in this field? It is recommended to dig deeper into the innovation points and highlight the innovative contributions.
2.As described in Methods, the literature search for this systematic review was based on PubMed database only, which might miss some important studies. It is suggested to expand the search scope and include more meaningful articles to make this manuscript more comprehensive.
3.What's the reason for this review to define the age of paediatric/childhood cancer between 0 to 19 years? I noticed that some systematic reviews defined this age range below 25 years and divided it into three categories: children/childhood (0–13), adolescents (13–21) and young adults (21–25).
4.An analysis of strengths and limitations of this study should be complemented.
5.Some outdated references should be replaced.
Reviewer 5 Report
Dear Authors
I have some questions and comments.
1. Introduction
It is important to assess who informs and who should inform patients and parents about methods and possibilities of fertility preservation?
2. Methods
what is Rayyan?
3. Discussion
- it is interesting, who pays for oncofertility in children in diffrent countries ?
-what are the risks of gonadotoxicity in CCS depending on the kind of diagnosis and treatment?
The authors should compare data about oncofertility in diffrent countries.
Round 2
Reviewer 1 Report
The authors have been very responsive to reviewer suggestions and the manuscript has been strengthened. Thank you for publishing in this important area.
Reviewer 4 Report
The authors have made sufficient modifications according to the comments, and I suggest that this paper be accepted without further modification.